# The Pain at Work Toolkit for Employees with Chronic or Persistent Pain: A Collaborative-Participatory Study

**DOI:** 10.3390/healthcare10010056

**Published:** 2021-12-29

**Authors:** Holly Blake, Sarah Somerset, Sarah Greaves

**Affiliations:** 1School of Health Sciences, University of Nottingham, Nottingham NG7 2HA, UK; sarah.somerset@nottingham.ac.uk (S.S.); sarah.greaves@nottingham.ac.uk (S.G.); 2NIHR Nottingham Biomedical Research Centre, Nottingham NG7 2UH, UK

**Keywords:** chronic pain, self-management, toolkit, participatory design, inclusion, workforce, workplace, occupational health, digital, technology

## Abstract

Self-management tools for people with chronic or persistent pain tend to focus on symptom reporting, treatment programmes or exercise and do not address barriers to work, facilitators of work ability, or workplace pain self-management strategies. We developed the Pain at Work (PAW) toolkit, an evidence-based digital toolkit to provide advice on how employees can self-manage their pain at work. In a collaborative-participatory design, 4-step Agile methodology (*N* = 452) was used to co-create the toolkit with healthcare professionals, employers and people with chronic or persistent pain. Step 1: stakeholder consultation event (*n* = 27) established content and format; Step 2: online survey with employees who have persistent pain (*n* = 274) showed employees fear disclosing their condition, and commonly report discrimination and lack of line manager support. Step 3: online employer survey (*n* = 107) showed employers rarely provide self-management materials or education around managing pain at work, occupational health recommendations for reasonable adjustments are not always actioned, and pain-related stigma is common. Step 4: Toolkit development integrated findings and recommendations from Steps 1–3, and iterative expert peer review was conducted (*n* = 40). The PAW toolkit provides (a) evidence-based guidelines and signposting around work-capacity advice and support; (b) self-management strategies around working with chronic or persistent pain, (c) promotion of healthy lifestyles, and quality of life at work; (d) advice on adjustments to working environments and workplace solutions to facilitate work participation.

## 1. Introduction

Chronic or persistent pain affects between one-third and one-half of the population of the United Kingdom (UK), corresponding to just under 28 million adults [1]. There is unequivocal evidence that chronic pain reduces quality of life [2]. This negative impact on quality of life for people living with pain has been exacerbated by the recent coronavirus disease (COVID-19) outbreak due to changes in work-related activities, and physical inactivity during periods of confinement [3]. COVID-19 is caused by severe acute respiratory syndrome coronavirus 2 (SARS-CoV-2) and was declared a pandemic in the UK by the World Health Organization (WHO) in March 2020. This pandemic has generated additional healthcare concerns for people with chronic pain due to postponed or cancelled elective surgical procedures, outpatient procedures and patient visits (including pain management services), resulting in delayed treatments, lack of continuity of care, and concerns about access to healthcare services and pain medications [4,5].

Chronic pain may impact on an individual’s ability to work [6,7,8]. Retention of people with chronic pain in the active workforce is important, since continued employment status is inversely related to pain severity [9], and extended absence from work impacts negatively on physical and mental health, as well as the economy [10,11,12]. Further, there is no conclusive evidence to support any specific tertiary-level (remedial) return to work intervention for workers with chronic pain who are already off sick [13], although it has been raised that managing pain, managing work relationships and making workplace adjustments are essential (but not straightforward) areas of a successful return for those with pain-related absence [14].

Chronic pain is considered a work-related stressor, since pain interference during the working day can impact on mood and lead to exhaustion, irrespective of pain severity [15]. For people with painful conditions, there are many employment challenges including physical limitations and ergonomics, work transitions or accommodations, stigma, the complexity of disclosure, social support at work, and the unpredictability of painful conditions or condition progression [16,17]. These are exacerbated by work-related factors such as high job demands (workload), low job control and decision authority, and low job support [18,19]. For employers, a particular challenge is not only absenteeism associated with pain [20], but the potential loss of productivity in people with chronic pain because of reduced ability at work (known as presenteeism) [21], which generates a financial burden [22,23].

There is a clear need for intervention to support people with chronic or persistent pain in the management of their condition at work for the benefit of individuals and organisations. Many organisations (particularly small to medium organisations) lack access to occupational health services for their employees. Those with access to services may lack knowledge about the impacts of chronic or persistent pain on working life. It has been proposed that occupational health and safety professionals do not necessarily have awareness, knowledge or training about chronic pain despite having involvement in the workplace management of chronic pain cases [24]. The provision of education and supportive materials for employees with chronic pain is therefore inconsistent or lacking across organisations and sectors. While educational intervention improves pain and disability in people with chronic pain of any aetiology [25,26,27], ‘patient’ targeted education tends to focus on neurophysiological aspects of pain and function and is less likely to focus on the management of the impacts of pain on working life. There is a current lack of interventions designed to assist with work-related chronic pain issues.

Further, workplace interventions for people with chronic pain tend to focus on the delivery of single approaches to self-management (e.g., exercise: [28]; strength training: [29,30]; mindfulness: [31,32,33]) and/or focus on specific painful conditions (e.g., neck pain: [34,35]; back pain: [36,37]) rather than offering comprehensive advice across a range of areas, which is suitable for employees with any pain condition. Digital technologies are increasingly used to support the self-management of chronic or persistent pain in various modalities (e.g., e-Health, m-Health, virtual reality: [38]; web-based: [39]) and have been used to improve pain or disability in a range of conditions (e.g., chronic low back pain: [40,41]; persistent musculoskeletal pain: [42]; headache: [43]. Digital interventions have potential for wide geographical reach and flexibility of access, which is particularly important given the increase in remote working and/or changes in working patterns that have occurred for many employees during (and likely following) the COVID-19 pandemic. However, existing digital interventions for people with chronic or persistent pain do not tend to address barriers to work, facilitators of work ability, or pain self-management in the context of work.

The aim of the study was to develop an evidence-based online toolkit to provide advice on how employees with any chronic or persistent pain condition can self-manage their condition at work. Toolkit development involved co-creation activities together with an interdisciplinary stakeholder group and expert review panel with members from the public, private and third sector. To achieve the aim, the objectives of the study were to (i) consult with a wider range of stakeholders to establish content and format of the toolkit; (ii) identify employer provisions and challenges relating to supporting employees with chronic or persistent pain; (iii) identify key challenges and support needs of employees with chronic or persistent pain; (iv) conduct iterative expert peer review to complete co-creation of a final toolkit which would be appropriate for use by any employee across all organisation types and size.

## 2. Methods

Rigorous development processes and engagement of stakeholders is essential for development of a high-quality intervention. In a collaborative-participatory design [44], we used an Agile Methodology approach as used in other published evaluations of workplace digital interventions [45,46,47], to develop a digital intervention to support people at work with chronic or persistent pain. The study took place at a higher education institution in England. Development followed a 4-step process (*N* = 450, Figure 1): (Step 1) a stakeholder consultation event (*n* = 27); (Step 2) an online survey with employees who have persistent pain (*n* = 274, 18–65 years); (Step 3) an online survey with employers (*n* = 107; 45 SMEs (Small to Medium Enterprises), 62 large organisations); (Step 4) toolkit development with iterative expert peer review (*n* = 40, 13M; 27F). The 4 steps involved stakeholders from academia, healthcare and industry, as well as people with lived experience of chronic or persistent pain. Consultation activities and online survey questions were developed by the research team and were intended to inform toolkit development. Our Agile approach utilised principles of Kanban methodology [48] in which steps 1–3 produced lists of toolkit and development tasks (allowing us to draw from a backlog) and the product (the Pain at Work (PAW) toolkit) was released to reviewers with each update, enabling iterative review. The description of the toolkit aligns with the Template for Intervention Description and Replication (TIDieR) Checklist [49] (Appendix A). The project team had expertise in participatory approaches for digital intervention development and Agile methodology. The study was conducted according to the guidelines of the Declaration of Helsinki and was classed as educational development and evaluation by the Research Ethics Committee of the University of Nottingham Faculty of Medicine and Health Sciences (Ref: FMHS 358-0921).

### 2.1. Step 1: Stakeholder Consultation Event

A two-hour face-to-face consultation workshop was held in February 2020, in a university medical school based within an acute hospital trust in England. Participants were identified through professional networks and purposively invited to include people with a wide range of chronic or persistent pain conditions, healthcare professionals, academics, employment advisors (including HR and occupational health), welfare and wellbeing officers, local councils, line managers, trade unions and workplace health champions. The event included a 15-min presentation delivered by a health psychologist on ‘Chronic or persistent pain at work’, including the project rationale and methodology. The facilitator had subject expertise, and prior experience of participatory design and Agile methodology. This was followed by three group activities that were based on discussion around (i) determining which groups should be consulted with regards content development and/or package dissemination, (ii) routes to technical and content development support, (iii) what format the package should take. Finally, the group were asked to review draft items developed by the project team for a brief employee and employer survey.

### 2.2. Step 2: Employee Survey

An online employee survey was distributed between March–April 2020, via public notices, pain charities and social media. This coincided with WHO’s declaration of COVID-19 as a pandemic in March 2020, which was followed by major pandemic-related impacts for the global workforce including a transition to remote working for a significant proportion of working-age adults. A repeat survey was released in October 2020 and left open for four months to maximise response rate (during the continuing pandemic) and capture additional workforce challenges that may have arisen during this unprecedented time. The survey platform was Jisc Online Surveys, a platform selected that meets UK accessibility requirements and is compliant with UK data protection laws. Data were collected on age, gender and employment status (employed, long-term absence from work, unemployed, retired, student). Employees were then asked to respond to three open-ended questions: 

Question 1: ‘In your line of work, as someone with chronic or persistent pain, what are the biggest challenges for you in meeting your work requirements?’,

Question 2: ‘What could be done to make your work situation better? (or alternatively if you are fully satisfied, what has been done to ensure your work situation meets your needs)’,

Question 3: ‘How could/are these changes at work best provided?’.

Free-text comments were analysed using thematic analysis employing an inductive approach in which coding and theme development were driven by the content of the comments. This involved analysis of semantic content of the entire free-text data and whether comments were of a positive or negative nature. A coding taxonomy was produced for sorting qualitative data into categories of participant experience, including concerns and challenges, impacts of a global pandemic on work and pain, and employee solutions.

### 2.3. Step 3: Employer Survey

An online employer survey was set up using the same platform and distributed in March–April 2020, via professional networks, and social media. Data were collected on organisation size, type and sector and the primary job role of the responding individual. Participants were asked to select which supports were in place at their organisation for people who have chronic or persistent pain, such as: policies (e.g., disability), pain self-management support, education or training sessions, employee assistance programme (EAP), counselling service, mental health awareness, physical exercise, physiotherapy, occupational health service, health check-ups or medical assessments, or private healthcare. Employers were then asked to respond to three open-ended questions:

Question 1: ‘In your line of work, what are the main barriers to meeting work requirements for people with chronic or persistent pain?’,

Question 2: ‘In your line of work, what are the main ways to support people with chronic or persistent pain?’,

Question 3: ‘Can you recommend any useful resources or materials that we can use to support people with chronic or persistent pain at work?’.

Analysis of free text comments was undertaken using thematic analysis as reported in Section 2.2. For employers, comments were coded into concerns and challenges, and employer solutions.

### 2.4. Step 4: Toolkit Development and Expert Peer Review

Development processes included content development, a virtual peer review panel, and technical development. These processes were all undertaken virtually due to the timing of the study which occurred during the global coronavirus disease (COVID-19) pandemic. Content was compiled by the project team, which included a psychologist with expertise in health and work, a welfare officer, and a workplace health researcher. The initial draft was informed by the stakeholder consultation and findings from the employee and employer surveys. A prototype toolkit was developed as an interactive portable document format (PDF). This was considered to be the ‘minimal viable product (MVP)’ [48] and was used in the agile development process. Forty people engaged in an expert peer review panel representing academic, health care, industry and community partners. Peer reviewers were purposively selected through direct approaches from the project team to relevant departments and individuals with appropriate professional expertise, and invitation sent via pain networks, pain charities supporting people with a range of chronic pain conditions and Pain Centre Versus Arthritis, a national pain research centre based at the University of Nottingham. Panel members reviewed the toolkit content and provided feedback using a peer review form containing items adapted from the HELM Open RLO-CETL Evaluation Toolkit for Reusable Learning Objects and Deployment of E-Learning Resources (Appendix A). Adopting an agile approach allowed for the provision of an MVP each time a reviewer was asked for feedback [50], e.g., after verbal or written feedback from a reviewer, revisions were made, and an updated version of the toolkit was sent to the next reviewers. The peer review form contained 10 question items, including consideration of pedagogy, format, usability, navigation, interactivity, delivery, ease of updating, distribution, and access [51]. Once peer review of content was complete, the PAW toolkit materials were transferred into Xerte, an open-source software for authoring learning objects that does not require any specific technical or programme skills. Xerte is free to download from the Xerte Community website [52].

## 3. Results

### 3.1. Step 1: Stakeholder Consultation Event

Twenty-seven people attended the event (12M, 15F). Attendees had expertise in nursing, pain management, pharmacy, rehabilitation, occupational therapy, occupational health, physiotherapy, human resource management, trade unions, health psychology, public health, workplace wellbeing or welfare, equality and inclusion, ergonomics, human factors, digital health or educational interventions, and/or lived experience of pain. Most attendees had expertise spanning multiple relevant areas, and attendees overall had expertise across the most common chronic pain conditions (i.e., back pain, arthritis, complex regional pain syndrome, multiple sclerosis, neuropathy, irritable bowel, headache and migraine, fibromyalgia, diabetes or cancer-related pain). Efforts were made to recruit a diverse group of attendees, including individuals of different age, gender, educational level, occupation and pain-related experience, nationality, and ethnic/cultural group. There was a consensus that awareness raising was required about chronic or persistent pain in the workplace setting, and that the toolkit would be a useful resource for people with chronic or persistent pain, irrespective of their specific condition. Suggestions were made for content and presentation of the PAW toolkit. Discussions led to the generation of a list of routes to existing materials and informed the development of a dissemination plan for the developed toolkit. It was considered that either a website, or a digital package would be a suitable platform for delivery. Draft items for the employee and employer surveys were reviewed and finalised. Stakeholders proposed the separation of identifiers from responses in the study analysis and reporting due to the perceived sensitivity of the subject area. 

### 3.2. Step 2: Employee Survey

#### 3.2.1. Employee Characteristics

Employees from all age categories responded (18–65+ years), although most responses came from those aged 35–64 years, with the highest proportion of respondents being 45–54 years. This was expected given that most of our survey respondents were employed (94.6%) and the employment rate in the UK is highest for 35–45 years [53]; also, the prevalence of moderate-severely disabling chronic pain increases with age [1]. Characteristics of employee respondents are shown in Table 1.

#### 3.2.2. Employee Concerns and Challenges

Challenges raised by people with pain are shown in Table 2. Employees reported many pain-related concerns that impacted on their work (e.g., intensity and duration of pain, unpredictability of pain and flare-ups, fatigue, sleep loss, medication). The majority of participants reported that pain and associated tiredness and fatigue had significant adverse effects on their perceived cognitive function (e.g., attention, memory, concentration or ability to focus) which they felt were important barriers to work ability. Employees raised many job-related barriers (e.g., heavy workloads, long hours, inflexible job role expectations, rigidity of work patterns). For some, flexible working hours were perceived to be potential helpful, but not sanctioned within their job role.

“*Inability to have flexible working to help me manage my health and still do my role, not everyone needs to be in the office every day*”.

Environmental issues were reported (e.g., requirement for work-related travel, mobility within and between sites, shared office space impacting on comfort and control over temperature regulation), as well as ergonomic challenges (e.g., lack of access to specialist equipment, the need for physical adjustments that were not possible, and an inability to establish comfortable positions). Shared workspaces and hot-desking presented particular challenges, as well as a requirement to attend meetings in other settings:

“*I share a workspace, so the fact that I have to adjust the chair, computer etc every shift, rather than having a permanent set up, can be difficult*”.

“*Going to meetings when I can’t have the right support. I need back support and a footrest then I’m fine, but meeting rooms usually have awful chairs-no back support and impossible to reach the floor. I also couldn’t hot desk as I need too many personal accessories, like a back rest, special mouse and keyboard, footrest*”.

Employees viewed their unaddressed issues as having negative impacts on work-related factors, such as an impaired working speed on days with higher pain levels, resulting in concerns about being perceived to have reduced performance and lower productivity.

“*I got pain and swelling in my wrists which can make typing and scrolling on computers difficult. I often have to stop writing which delays my work*”.

This led to significant impacts to employee wellbeing, and participants reported low mood, fear, anxiety, lack of work-life balance, and overall poor quality of life. 

“*Continuing to meet my daily obligations, no matter how much pain I am in that day or that week. Hard sometimes to keep up a cheery exterior*”.

“*Not having a work life balance-putting too much energy into work and not leaving energy for fun or anything else…trying not to show how hard I work to prove I can still do it*”.

While some concerns were specific to health conditions and not possible to address, many of the challenges reported were seen to be amenable to change with appropriate employee or employer education and training, or provision of reasonable adjustments. However, the most commonly raised concern was related to the variability of chronic or persistent pain with regards to pain levels, and work capability on any given day:

“*The sporadic nature of living with a long-term health condition… I can have periods of months with few problems and then a flare up which requires hospital treatment*”.

“*Some days can be much worse in terms of pain and the impact is greater*”.

“*Each day is changeable. I might be having a good day and able to meet some of the more physical elements of my role one day, then the next be struggling even with less active tasks*”.

Employees referred to the challenges of hidden disabilities in the workplace, and prevailing negative attitudes of managers and peers towards something they could not see or understand. Employees alluded to a perceived stigma towards people with pain which impacted negatively on quality of working life, career progression opportunities and ultimately mental wellbeing of employees. 

“*Others not understanding that because you don’t ‘look in pain’ you must be a whinger or making it up. Fear if you admit how much pain you are in, they will fire you or retire you or you won’t get promotions*”.

“*Having a suitable ergonomic desk and chair has made a massive difference, but it has been challenging when colleagues have not appreciated how necessary or important these items are to me*”.

Employees referred to line managers and colleagues lacking knowledge about pain (or any disability) and its work-related impacts or refusing to make adjustments.

“*I work in large open plan office and if you ask for adjustments the other people in the office will complain and senior management will blame me for it*”.

“*I feel I have to apologise for being in pain. My job does not get adapted and I feel that if I make too much of an issue I will lose my employment, especially in these times*”.

One employee referred to the “*challenges of explaining how difficult and draining it can feel to my manager, who is well-meaning but doesn’t really get it*”. For others, managers were perceived to be less well-meaning, and reference was made to a lack of compassion and support towards people with chronic or persistent pain which engendered feelings of workplace inequality. There was a “*fear of being labelled*” and a “*fear of not being understood*” coupled with uncertainty around policies and workplace practices to support staff with pain:

“*[there is a] lack of clarity around how pain related sick leave would be managed in line with current HR policy*”.

“*Sickness absence policy isn’t set up for people with chronic or persistent pain*”.

#### 3.2.3. Impacts of a Global Pandemic on Work and Pain 

The pain-related changes brought about by changes in the nature of work during the global COVID-19 pandemic were reported by participants (Table 2). Participants reported changes in their work circumstances during this time, for example, being furloughed, or transitioning from office-based to home-based working. Those who had adjustments to their workstations in the workplace setting reported significant difficulties with a transition to the home setting where these adjustments may not be available, or where the space was not well established for home working.

“*At work I have access to a standing desk and ergonomic chair. Since the pandemic hit, I’m working from home all the time, but I have no access to those*”.

Some employees reported working in unsuitable environments at home during the pandemic, with poor workstations and cramped spaces leading to increases in pain. Some reported that reasonable adjustments had been made by employers in response to COVID-19 and the resulting transition to home-working but that this had not been helpful since the workloads remained the same even when hours were reduced.

“*Currently, I am on slightly reduced hours as I don’t have all my reasonable adjustments at home, but I am still completing the same work as I would in full-time hours*”.

Reports of high stress and anxiety related to the pandemic were challenging for some, with regards their ability to relax, in order to self-manage their pain. Participants were asked only to report on the main challenges for self-management of their pain at work. However, without being prompted, it is notable that many participants chose to raise several positive aspects of a transition to remote working, as a result of COVID-19. Some reported that this transition had aided self-management and led to a significant reduction in pain levels.

“*Recent changes for COVID-19 mean I am now full time from home and coping extremely well*”.

“*My health has improved massively since lock down as I can start later and work later on days when pain and stiffness are too hard in the morning*”.

This was perceived to be related to the removal of the commute from home to work (or even movement around and between sites) coupled with an ability to take breaks at times to suit them. Participants felt they had more control over their work patterns. Some reported that working at home had allowed them to wear more comfortable clothing and sit in more comfortable positions when working which had helped them to self-manage their pain. Home working was generally seen to be a positive.

“*I work from home and do not work set hours, which means I can work around my pain, even working from bed when necessary*”.

#### 3.2.4. Employee Solutions

There was a general consensus that flexible working patterns and personal workspaces (for office-based staff) that could be tailored to suit the employees’ needs were both important for the self-management of pain at work. However, it was clear that this is not uniformly provided in all job roles. Employees advocated the potential benefits of occupational health assessments, which could result in actions such as flexible working, ergonomic assessment and appropriate adjustments to workstations, equipment, and job requirements. However, this was only viewed to be successful if properly managed with timely appointments, follow-up, and action. Acting on recommendations for reasonable adjustments was seen to be essential to avoid a perception that inclusion initiatives, disability or health and safety policies were ‘box ticking exercises’ or ‘lip service’. It was recommended that risk assessments should be included in staff inductions to identify disability-related needs and deliver early intervention; workplace disability interventions should not be dependent upon ownership of a blue badge (UK local government disabled parking permit) to class an employee as disabled and eligible for employer support.

Ultimately, it was proposed that organisations should actively raise awareness around chronic pain and disability more broadly. It was proposed that line managers should be trained not only in reasonable adjustments, but also the “*practical everyday demands of the role*” for someone living with pain as well as the unpredictable nature of pain.

“*The only thing which would make a work situation better is for there to be better understanding and training that chronic or persistent pain conditions are invisible and fluctuating. That would avoid some difficult conversations*”.

“*Educate people that not all disabilities are visible and that many of us live with conditions which are life limiting. It’s not our choice*”.

The majority of participants referred to communication from ‘gatekeepers’ as being key; it was suggested that line managers should be trained in how to communicate appropriately with staff (both privately and publicly) with “*more empathy and understanding*” in cases where reasonable adjustments are required. The most commonly reported and most challenging barrier for participants appeared to be judgmental attitudes and insensitive comments from line managers and team members. Addressing this was seen to be essential to foster an organisational culture of kindness and compassion, reduce stigma relating to people with disabilities, and avoid unnecessary psychological harm imparted by others in the workplace (e.g., embarrassment, fear, anxiety, shame, and guilt).

“*I have been provided with adapted workspace but was reminded of this repeatedly at staff meetings that it had eaten into budget*”.

“*Emotional support, reasonable adjustments, open environment to talk about it without me feeling embarrassed intimidated or uncomfortable or that I’m not pulling my weight or that I can’t do my job*”.

It was seen to be essential to update policies to support people with chronic or persistent pain at work and that structural organisational changes would be required to implement them. For example, participants highlighted that disability should be included in workload models, taken into account in performance reviews, and financial support for adjustments should come from a central budget not devolved to local departments which was seen to result in inequity. Employees preferred not to be simply directed to a website for information but advocated the importance of a named person they could speak to about health and work concerns, in addition to an informative disability ‘hub’, disability-related mentorship schemes and access to positive stories from people with lived experience of pain at work. Due to the impacts of persistent pain on work and personal life, it was thought that mental health and wellbeing support was important, together with tools to support self-management of pain. Employees recommended helpful apps, and the provision of information about healthy lifestyles (e.g., diet, physical activity, stretching), physical therapy, massage, work pacing, managing fatigue, mindfulness and other coping strategies.

### 3.3. Step 3: Employer Survey

#### 3.3.1. Employer Characteristics

One hundred and seven employer representatives from the public, private and third sector completed the survey (45 small and medium-sized enterprises, 62 large organisations). Employers represented all sectors, but the highest number of responses came from construction, tourism, healthcare, and education. The majority of employers did not provide any form of self-management materials (86%) or education (92.5%) for people with chronic or persistent pain. Almost half reported that their organisation had no policies in place to support people in the workplace with chronic or persistent pain (e.g., disability policy) (45.8%). Almost half the employers reported that there was no support for mental health (49.5%), although counselling services and employee assistance programmes were available in some organisations. Only a minority provided physiotherapy (11.2%) or promoted physical exercise (15%). Seventy-two per cent of organisations provided access to occupational health services, a quarter offered medical assessments or health check-ups (25.2%), and just over a third of organisations offered access to private healthcare (27.1%). SMEs were significantly less likely to have support available for their employees than large organisations, in every area. For employer characteristics and provisions by organisation size, see Table 3.

#### 3.3.2. Employer Concerns and Challenges

Many of the employers recognised that there were job-related challenges for employees with chronic or persistent pain (Table 4). In particular, they referred to the impacts of heavy workloads, manual handling, peripatetic working practices (e.g., working in multiple areas, transport of equipment or resources from room to room) and inflexible work patterns. Some of the employers reported the benefits of occupational health services for employees, although not all organisations offered these services and when they were offered, there was a consensus that referral processes often took too long, and processes needed to be streamlined.

In alignment with employee views, employers also reported the presence of a high level of stigma in the workplace around chronic or persistent pain, and disability. This was viewed to negatively impact on work culture and employee wellbeing and was attributed to a lack of understanding (of managers and the workforce more broadly) about the nature of chronic or persistent pain and its impacts at work.

“*I am a medical doctor…before I had chronic or persistent pain it never occurred to me that there is so much ignorance and prejudice about chronic or persistent pain*”.

Employers repeatedly alluded to the negative attitudes of line managers in their organisations towards employees with painful conditions. They reported that dismissive comments were often made, and prohibitive management behaviours delayed or prevented appropriate support being put into place. Employers admitted that line managers would often fail to act on occupational health recommendations, or they would actively challenge employees and occupational health assessors on the outcome of assessments.

“*…Line managers ignoring the recommendations from the occupational health team…Delays in the approval and delivery of disability equipment*”.

“*‘anecdotal comments-Does she need a chair when a cushion will do?’ or ‘If she can’t drive long distances what’s wrong with stopping to stretch her legs?’*”

For employers, one of the main challenges was managing pain-related sickness absence and presenteeism (being present at work when unwell). With relation to sickness absence, concerns were raised about striking a balance between supporting the employee by recommending sick leave, then organising cover for sickness absenteeism, while avoiding their employees being subject to disciplinary measures for repeated absences. With regards presenteeism, employers reported that their staff members would come to work with other non-related sickness (including communicable conditions), particularly if workloads were heavy, to avoid getting a poor absence record.

“*In a small company, the hardest thing is cover if someone is absent from work. People don’t tend to take sick days as they don’t want to let people down*”.

#### 3.3.3. Employer Solutions

Employers believed that disclosure of chronic conditions was important and helpful, although it was recognised that there were various barriers to disclosure for employees. They perceived the barriers to disclosure primarily relating to negative workplace culture, lack of managerial support, and non-recognition of certain conditions as a disability under the Equality Act 2010. Employers highlighted the importance of a psychologically safe working environment in facilitating disclosure and communication with employees about their needs.

“*I thought we didn’t have anyone with chronic or persistent pain, but this did make me think about types of pain that are less visible, and as a leader it’s about being sensitive to that and knowing that people might be struggling but might not disclose it, in case they let you down. So, the biggest challenge is probably feeling safe to disclose health issues*”.

Employers proposed various strategies for support, including appropriate policies, and job-related adjustments (e.g., flexible work hours, adaptions, increased use of video-conferencing, 45-min cap on meeting length) as well as supportive services (e.g., support schemes such as Access to Work in the UK, disability champions, peer support networks, occupational health services, private healthcare, counselling and employee assistance programmes) and promotion of self-management approaches (e.g., physical therapy, relaxation, physical and emotional wellbeing). Employers advocated for line manager training to improve their understanding about the impacts and limitations of pain, as well as possible adjustments. It was perceived that education and training may reduce the prevalence of derogatory comments and discrimination in the workplace. It was acknowledged that employers should be actively promoting self-management of chronic or persistent pain to ensure that people with pain can be as productive as possible at work. However, it was recognised that employers tended to focus only on policies and meeting legal requirements and so promotion of self-management did not generally occur in the workplace. Proposed approaches to support employees with chronic or persistent pain arising from employee and employer surveys are further discussed in Section 4.

### 3.4. Step 4 Part 1: Virtual Peer Review Panel

All 40 reviewers (100%) engaged with this process and provided reviewer feedback (Figure 2). This collaborative-participatory activity focused on toolkit design, usability and human factors. Suggestions for change primarily related to language and terminology, revisions to the contents page, and further signposting to resources and self-management apps, as well as minor issues of presentation and consistency. All revisions were made iteratively into the MVP, and the content was then transferred into a Xerte [52] learning package hosted on a secure server at a university.

### 3.5. Step 4 Part 2: The PAW Toolkit 

The developed toolkit is publicly accessible [54] and highly flexible to user needs and preferences, meaning the way it is used can be highly personalised, it can be accessed at any time, in any location and worked through at any pace. The ‘pre-planned schedule’ consists of five sections which is considered the full ‘dose’ of intervention content, designed to be accessed in succession from Section 1, Section 2, Section 3, Section 4 and Section 5. However, users can choose the ‘actual schedule’ (the order of sections visited or re-visited), the ‘dose’ they receive (how much of the content they access), the ‘duration’ (how long they access it for), and the ‘intensity’ (how often they access it) of the intervention. The toolkit provides evidence-based guidelines and signposting in five broad areas outlined in Figure 3. An example of presentation style is provided in Figure 4. Content and presentation were developed consider known enablers and barriers to engagement in digital interventions for people with chronic pain, through flexibility for access, inclusivity for people with disabilities, and low technological skill requirement (e.g., [55]).

## 4. Discussion

To our knowledge, the PAW toolkit is the first accessible, digital resource to support employees at work who have chronic or persistent pain. It is publicly accessible, free to use and was developed through a rigorous, participatory design process involving surveys, consultations and peer review, engaging employees who live with chronic or persistent pain, employers and stakeholders with expertise in workplace issues and/or the management of pain. The toolkit is perceived to be relevant to employees from any size or type of organisation and addresses a clear need identified through review of evidence, stakeholder consultation and surveys with employees and employers.

The development process highlighted key concerns and challenges experienced by people with pain that were recognised by employers. Many of their concerns could be modifiable with intervention or appropriate support. First, there is a clear need for awareness raising in the workplace regarding the experience of pain for some employees, their pain-related concerns, the impacts of pain on physical and mental wellbeing, and quality of working life (as well as leisure time activities). Our study highlights the need to raise awareness among managers and the working population. However, prior research has also shown the importance of continuing education on chronic pain amongst occupational safety and health professionals [24] to whom employees might turn for advice. The negative impact of chronic pain on the quality of social and working lives is already well established [2,6,7,8]. Our findings suggest that lack of knowledge and awareness of pain and disability among employers markedly increases the inequity experienced by people working with chronic pain in terms of work experiences, opportunities, career progression and work-life balance. Such issues have been discussed elsewhere [56,57,58].

Personal adjustments and workplace interventions are reported to be important determinants for staying at work [59]. However, employees are not always aware of the support they can access or approaches to help them self-manage a painful condition. The PAW toolkit addresses this gap with provision of information and supportive resources. With relation to self-management, we advocate promotion of physical and mental health at work in line with the call to action to increase awareness of effective nonpharmacologic treatments for pain [60]. These are wide-ranging, but examples include physical activity [61,62], exercise training [63], Tai Chi [64], yoga [65], pilates [66,67], nutrition and supplements [68], mindfulness meditation [31,32,69], massage therapy [70], acupuncture [71] and psychological therapies [72,73,74,75,76].

Concerns that are specific to job types or roles, or the working environment warrant discussion between employee and employer. Such concerns may be actionable but rely on disability disclosure and employee help-seeking. The PAW toolkit encourages disability disclosure and advocates help-seeking by empowering employees with knowledge about likely options and routes to support. Nevertheless, help-seeking and disability disclosure can be influenced by factors other than knowledge. There are socioeconomic, racial, ethnic and cultural discrepancies in pain beliefs, cognitions and behaviours that may impact on both disability disclosure and help-seeking behaviour [77], and this warrants further exploration.

Employees in our study reported a range of practical issues associated with specific job roles or environments, that could be addressed to some extent with appropriate assessment and intervention. For example, through modifications to activity, work or work patterns to accommodate difficulties related to function and pain while reducing activity “avoidance” (e.g., assistive technology, ergonomic assessment, regular breaks, change in scheduling, modified duties, flexible hours to attend rehabilitation or treatment). Some participants reported positive experiences of reasonable adjustments, and ergonomic assessment or occupational health referrals can be beneficial for employees with long-term conditions. However, our participants strongly advocated that occupational assessment is only meaningful to employees if recommendations are then acted upon and followed up in a timely way, which unfortunately was not common practice. Even more notable is the high proportion of employees that have no access to any form of workplace support. For example, SMEs constitute 99.9% of the UK business population [78] but smaller organisations often have less access to, or less advanced occupational health and safety services for their staff [79].

One of the most significant barriers for employees (which was recognised by employers), was a broad lack of compassion and understanding of line managers towards employees with chronic or persistent pain. While some managers struggled with managing the practical aspect of employees requiring support or periods of absence, others simply held negative attitudes towards employees with disabilities and made stigmatising or discriminatory remarks. This was perceived to be the most challenging aspect of working with long-term pain for many employees, since it fostered a negative workplace culture, and created a psychologically unsafe work environment. Psychological safety is a condition in which human beings feel (1) included, (2) safe to learn, (3) safe to contribute, and (4) safe to challenge the status quo–all without fear of being embarrassed, marginalised, or punished in some way [80]. Without psychological safety, disability disclosure is less likely. Paradoxically, employers completing our survey advocated for early disclosure, but the workplace culture and management behaviours often prohibited this. This aligns with prior research in which people with chronic pain were hesitant to disclose for fear of negative outcomes, yet employers indicate a preference for early disclosure [81].

Many participants who had disclosed their condition perceived that they were then discriminated against in the workplace, or their concerns were dismissed. Both employers and employees indicated that line managers often created obstacles for employees with long-term pain, ignored occupational health recommendations or even belittled employees for requests made. This not only has serious implications for work ability in people with chronic or persistent pain, but it exacerbates inequality in the workplace, and poses risks to career progression, as well as the physical and mental wellbeing of employees. A relationship between workplace discrimination and chronic pain has been identified previously [82,83]. In healthcare, the Institute of Medicine called for ‘a cultural transformation in the way in which pain is perceived, judged and treated’ [84] and this ethos should be transferable to the workplace setting; similarly, others advocate that stigmatisation of chronic pain, and its consequences, can only be addressed at a policy and practice (rather than individual cognitive) level [85].

Based on our findings through this rigorous PAW toolkit development process, we make recommendations for supporting employees with chronic or persistent pain (Figure 5). Many of these recommendations require input from the employer, such as disclosure, and seeking reasonable adjustments (e.g., line manager or work performance appraiser). Therefore, we particularly advocate for education and training of line managers to raise awareness about long-term pain and disability, reduce stigma for employees, enhance compassion and communication about pain-related needs in the workplace, and ultimately increase the likelihood of appropriate supports being put in place for individuals with pain. This is becoming increasingly pertinent in the light of the long-lasting global COVID-19 pandemic, during which new chronic pain, or exacerbations of pre-existing chronic pain is predicted [86], and the delivery of chronic pain care has been severely impacted [4,5,87].

### Study Limitations and Future Research

The online surveys were intentionally brief to maximise completion and provide only information that was required for the corresponding toolkit development step. We therefore collected details of age, gender and employment status (employees), and level of seniority, sector, and organisation type (employers). We did not collect data on the occupation, race/ethnicity or socio-economic status of employees completing surveys, or the occupation of employer representatives, although these factors may impact on perceptions of pain, help-seeking (or support-giving) behaviours. Specific beliefs about the nature of pain and disability play an important role; for example, a cultural group may hold expectations and acceptance of pain as part of normal life, and this may determine whether pain is viewed as a ‘problem’ that requires management and solutions. Some cultural groups may be resistant to disclosure of chronic conditions and help-seeking or may not accept individual responsibility for self-management of health conditions. Although an exploration of cultural influence on pain and self-management are beyond the scope of this study, there is a need to further explore the applicability of the PAW toolkit in different cultural groups and identify any potential cultural differences in pain perceptions, challenges and solutions that would allow for adaptation of the existing toolkit for different settings and contexts.

The PAW toolkit is intended for use by employees with chronic or persistent pain rather than managers or organisational leaders, although we recommend that employers familiarise themselves with the toolkit content to be aware of the self-management advice provided for employees and we reiterate the value of awareness-raising, education and training for both employees and line managers. It should be noted that through the collaborative-participatory approach taken, concerns of managers and organisational leaders were raised that the PAW toolkit (as an individual employee-level resource) cannot address. These not only included concerns of managers (e.g., processes, knowledge of supportive services) but also fundamental issues relating to workplace culture (e.g., the need to decrease pain-related stigma, and create a psychologically safe work environment to encourage disclose and help-seeking). There are many examples, but transferable knowledge from studies of stigma in healthcare settings suggests that stigma is best addressed by empowering individuals, targeting all levels and occupational groups, addressing multiple stigmas at once, and simultaneously targeting multiple ecological levels, such as targeting both individual attitudes and practices as well as policies and environment within the particular setting [88]. Vocational rehabilitation research focusing on specific conditions advocates that organisational responses to disclosure need to demonstrate trust and inclusive decision making, focus on employee ‘abilities’, and enhance perceptions of psychological safety at work (e.g., multiple sclerosis [89]).

Overall, individual-level interventions need to be delivered alongside organisational-level interventions to maximise the potential for positive outcomes. Explicitly, people who have chronic or persistent pain may do their best to gain the knowledge, confidence and skills required to self-manage, but effective self-management and help-seeking will only be possible with the right opportunities, support, and autonomy.

## 5. Conclusions

Efforts to support self-management of chronic or persistent pain are increasingly important, particularly due to the global work impacts of the COVID-19 pandemic. Employers do not currently routinely provide guidance or support for staff with chronic or persistent pain. The PAW toolkit is a new resource to support employees with managing chronic pain at work, co-created with healthcare professionals, employers, and people with persistent pain. The PAW toolkit can be widely implemented to support employees with chronic or persistent pain in the workplace. Disability policies alongside line manager education and training are recommended to foster a psychological safe work environment, maximise employee support and facilitate appropriate actions. Further research could explore the impact of the PAW toolkit on employee pain, wellbeing and support, and organisational outcomes.

## Figures and Tables

**Figure 1 healthcare-10-00056-f001:**
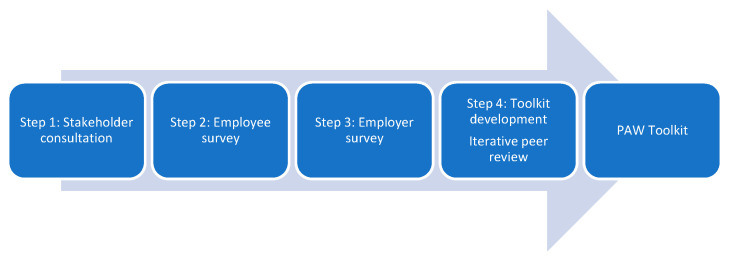
Four-step Agile development process.

**Figure 2 healthcare-10-00056-f002:**
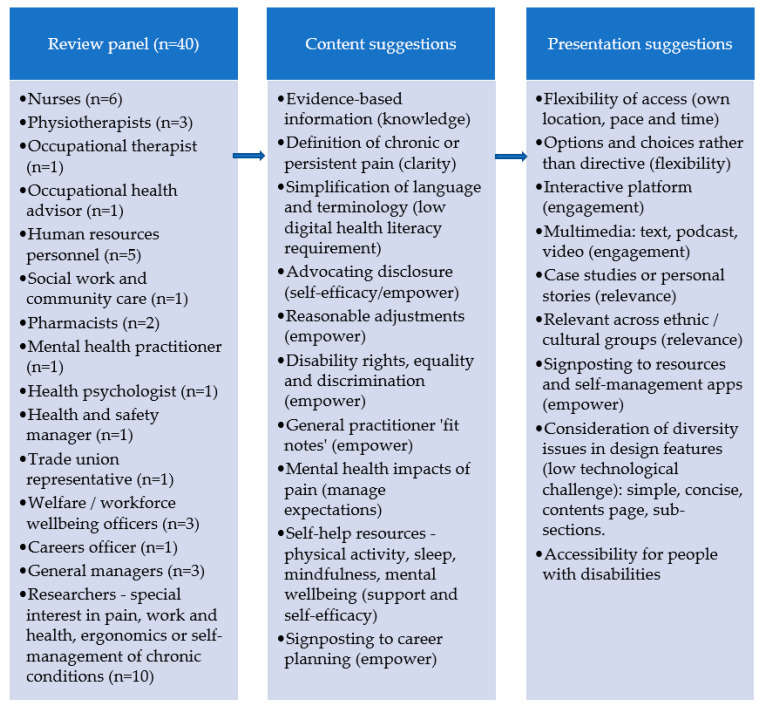
Examples from the peer review process.

**Figure 3 healthcare-10-00056-f003:**
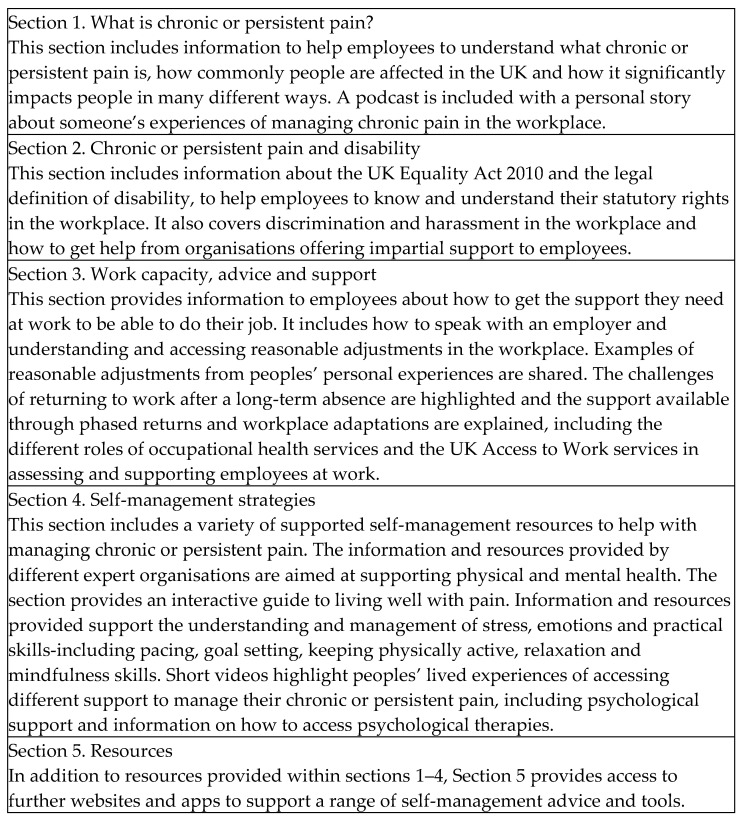
PAW Toolkit sections and content.

**Figure 4 healthcare-10-00056-f004:**
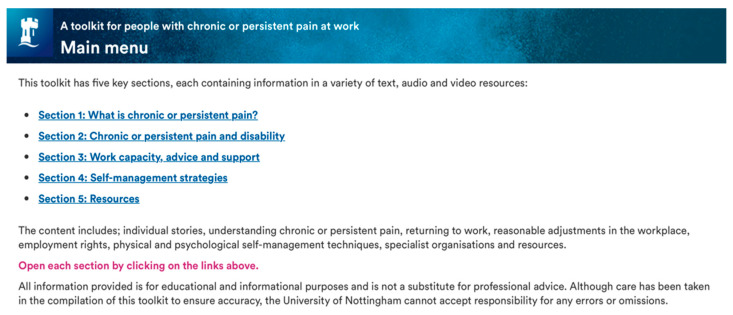
Example screenshot from the PAW Toolkit† (Version 1.0, 2021). (https://www.nottingham.ac.uk/toolkits/play_24452) (accessed on 17 November 2021). † see [54] to access toolkit, credits, embedded links and additional resources.

**Figure 5 healthcare-10-00056-f005:**
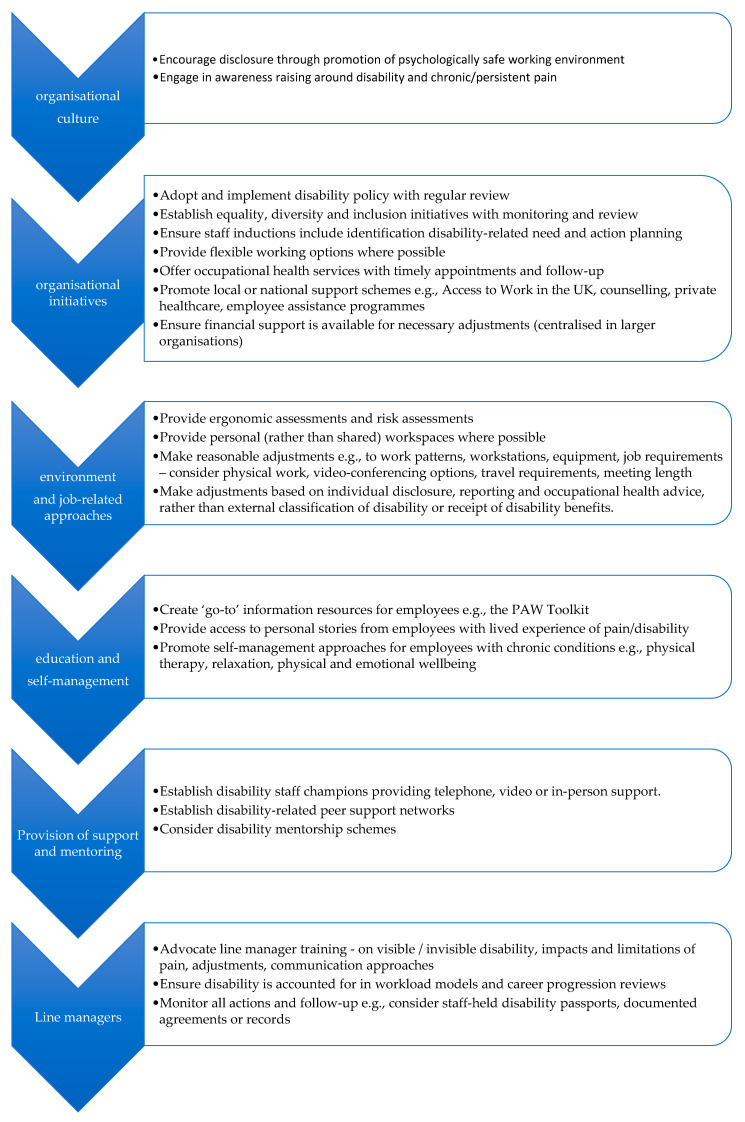
Proposed approaches to support employees with chronic or persistent pain.

**Table 1 healthcare-10-00056-t001:** Employee characteristics.

	Survey Participants (*n* = 274, 100%)			Total *n* = 274 (100%)
	Male *n* = 150 (54%)	Female *n* = 121 (43.5%)	Non-Binary *n* = 3 (2.5%)	
Age category (years)				
18–24	6 (4.0%)	5 (4.1%)	0 (%)	11 (4.0%)
25–34	24 (16.0%)	12 (10.0%)	0 (0.0%)	36 (13.1%)
35–44	35 (23.3%)	24 (19.8%)	3 (100.0%)	62 (22.6%)
45–54	53 (35.3%)	45 (37.2%)	0 (0.0%)	98 (35.8%)
55–64	31 (20.7%)	30 (24.8%)	0 (0.0%)	61 (22.3%)
65+	1 (0.7%)	5 (4.1%)	0 (0.0%)	6 (2.2%)
Work status	141 (94.0%)	115 (95.0%)	3 (100.0%)	263 (96.0%)
Employed (FT/PT)	3 (2.0%)	3 (2.5%)	0 (0.0%)	6 (2.2%)
Long-term absent from work	1 (0.7%)	1 (0.8%)	0 (0.0%)	2 (0.7%)
Unemployed	1 (0.7%)	0 (0.0%)	0 (0.0%)	1 (0.4%)
Retired	4 (2.7%)	2 (1.7%)	0 (0.0%)	6 (2.2%)
Other ^+^	141 (94.0%)	115 (95.0%)	3 (100.0%)	263 (96.0%)

^+^ Other: office or laboratory-based higher education research student.

**Table 2 healthcare-10-00056-t002:** Employee concerns and challenges.

Nature of Employee Concern
**Pain-related concerns**
Invisibility of pain (hidden disability) Medication and side effects (e.g., drowsinesss) Focus, concentration, memory Disturbed sleep Fatigue Low energy levels Physical fitness (as a safety concern)
**Emotional impacts**
Shame and guilt (related to ‘difference’, perceptions of special treatment or absence) Fear (related to job security, the need for absence) Pain-related anxiety (related to flare-ups) Job-related anxiety (related to job security) Low mood or depression
**Quality of life impacts**
Financial impacts (related to cost of treatment or equipment, loss of income if not offering a competitive service) Social isolation No work-life balance (related to using downtime to keep up with work because of the need to pace activities)
**Environmental concerns**
Prolonged sitting or standing Need to carry equipment Lack of car parking near to work area Meetings booked in different buildings Access to facilities (e.g., toilets/catering on different floor or in different building) Travel to work and between sites (walking or driving)
**Ergonomics**
Chairs-non-adjustable/no lumbar support No adaptions in meeting rooms Requirements for safety clothing (e.g., heavy shoes) Cramped workstations Shared workspaces and equipment (e.g., need to re-adjust daily, impacts of heat or air conditioning) Repetitive work Lack of access to desk assessment Lack of access to occupational health services
**Job-related concerns**
Non-disclosure (feeling unable) Workload-being unable to take breaks Length of activities (such as training sessions and meetings) Number of contracted or expected hours present at work (full days) Expectations of the job (e.g., required overnight stays) Time out to attend medical appointments Safety concerns with precision work (e.g., cutting, scoring, opening chemical bottles) Work quality impacts (related to concentration) Work productivity impacts (related to keeping to timelines and deadlines set by, or expected by others)
**Impact of line managers and peers**
Stigma and negative attitudes Hindered career development (related to stigma or reduced opportunities) Lack of understanding about pain impacts Lack of knowledge about support available Lack of compassion and unkindness Knowing who to talk to Knowing how to communicate about pain
**Impacts of COVID-19 ^+^**
*Negative impacts:* Long virtual meetings (related to increased use of technology) Inappropriate space for remote working (related to cramped, shared or noisy spaces) Lack of adaptions or required equipment at home *Positive impacts:* Increased flexibility in working hours More control over work patterns Increased comfort (related to clothing, and sitting positions) Reduction or removal of travel (related to commuting, expectations of overnight stays, travel between and within sites) Reduced medication (related to work flexibility, increased comfort and reduced travel)

Note: ^+^ The virus responsible for Coronavirus Disease (COVID-19) is severe acute respiratory syndrome coronavirus 2 (SARS-CoV-2).

**Table 3 healthcare-10-00056-t003:** Employer characteristics and provisions.

	Organisation Size ^+^ *n* = 107 (100%)				Total *n* = 107 (100%)
	Micro (*n* = 7)	Small (*n* = 16)	Medium (*n* = 22)	Large (*n* = 62)	
Primary job role					
Worker/employee	0 (0.0%)	1 (6.3%)	3 (13.6%)	15 (24.2%)	19 (17.8%)
Middle manager/team leader	2 (28.6%)	5 (31.3%)	14 (63.6%)	33 (53.2%)	54 (50.5%)
Senior manager/director/chief executive	5 (71.4%)	10 (62.5%)	5 (22.7%)	14 (22.6%)	34 (31.8%)
Sector					
Public	2 (28.6%)	4 (25.0%)	5 (22.7%)	34 (54.8%)	45 (42.1%)
Private	3 (42.9)	7 (43.7%)	14 (63.6%)	24 (38.7%)	48 (44.9%)
Third	2 (28.6%)	5 (31.3%)	3 (13.6%)	4 (6.5%)	14 (13.1%)
Organisation type					
Construction	0 (0.0%)	2 (12.5%)	7 (31.8%)	15 (24.2%)	24 (22.4%)
IT and internet	2 (28.6%)	1 (6.3%)	1 (4.5%)	1 (1.6%)	5 (4.7%)
Manufacturing and production	0 (0.0%)	2 (12.5%)	2 (9.0%)	0 (0.0%)	4 (3.7%)
Retail	1 (14.3%)	2 (12.5%)	0 (0.0%)	1 (1.6%)	4 (3.7%)
Tourism	1 (14.3%)	1 (6.3%)	2 (9.0%)	0 (0.0%)	4 (3.7%)
Education	2 (28.6%)	1 (6.3%)	3 (13.6%)	29 (46.8%)	35 (32.7%)
Healthcare	1 (14.3%)	0 (0.0%)	3 (13.6%)	9 (14.5%)	13 (12.1%)
Other	0 (0.0%)	7 (43.7%)	4 (18.2%)	7 (11.3%)	18 (16.8%)
Current provision					
Disability Policies					
Yes	3 (42.9)	11 (68.7%)	11 (50.0%)	33 (53.2%)	58 (54.2%)
No	4 (57.1%)	5 (31.3%)	11 (50.0%)	29 (46.8%)	49 (45.8%)
Self-management					
Yes	0 (0.0%)	2 (12.5%)	3 (13.6%)	10 (16.1%)	15 (14.0%)
No	7 (100.0%)	14 (87.5%)	19 (86.4%)	52 (83.9%)	92 (86.0%)
Education/training					
Yes	0 (0.0%)	1 (6.3%)	0 (0.0%)	7 (11.3%)	8 (7.5%)
No	7 (100.0%)	15 (93.7%)	22 (100.0%)	55 (88.7%)	99 (92.5%)
Counselling service					
Yes	2 (28.6%)	3 (18.8%)	5 (22.7%)	43 (69.4%)	53 (49.5%)
No	5 (71.4%)	13 (81.2%)	17 (77.3%)	19 (30.6%)	54 (50.5%)
Medical assessments					
Yes	0 (0.0%)	3 (18.8%)	6 (27.3%)	18 (29.0%)	27 (25.2%)
No	7 (100.0%)	13 (81.2%)	16 (72.7%)	44 (71.0%)	80 (74.8%)
Occupational health					
Yes	3 (42.9)	7 (43.7%)	15 (68.2%)	52 (83.9%)	77 (72.0%)
No	4 (57.1%)	9 (56.3%)	7 (31.8%)	10 (16.1%)	30 (28.0%)
Private healthcare					
Yes	0 (0.0%)	2 (12.5%)	9 (41.0%)	18 (29.0%)	29 (27.1%)
No	7 (100.0%)	14 (87.5%)	13 (59.0%)	44 (71.0%)	78 (72.9%)
EAP helpline					
Yes	0 (0.0%)	1 (6.3%)	5 (22.7%)	28 (45.2%)	34 (31.8%)
No	7 (100.0%)	15 (93.7%)	17 (77.3%)	34 (54.8%)	73 (68.2%)
Physical exercise					
Yes	1 (14.3%)	1 (6.3%)	0 (0.0%)	14 (22.6%)	16 (15.0%)
No	6 (85.7%)	15 (93.7%)	22 (100.0%)	48 (77.4%)	91 (85.0%)
Mental health					
Yes	3 (42.9)	2 (12.5%)	10 (45.5%)	39 (62.9%)	54 (50.5%)
No	4 (57.1)	14 (87.5%)	12 (54.5%)	23 (37.1%)	53 (49.5%)
Physiotherapy					
Yes	0 (0.0%)	0 (0.0%)	1 (4.5%)	11 (17.7%)	12 (11.2%)
No	7 (100.0%)	16 (100.0%)	21 (95.5%)	51 (82.3%)	95 (88.8%)

Note: EAP Employee assistance programme. ^+^ Organisation size: Micro 0–9 employees, Small 10–49 employees, Medium 50–249 employees, Large >249 employees.

**Table 4 healthcare-10-00056-t004:** Employer concerns and challenges.

Nature of Employer Concern
**Organisational concerns**
Covering staff absence in micro and small organisations Access to funds for adaptions/support (particularly SMEs **^+^**) Absence of OH provision or lengthy referral processes for OH services Challenges of managing sickness absence Balancing risks of presenteeism (particularly risks of communicable illness) Policy concerns around absence records/disciplinary measures for repeated absences Equality Act and employee statutory rights
**Job-related concerns**
Managing heavy workloads Requirement for manual handling Peripatetic working practices Inflexible work patterns
**Impact of line managers and peers**
High level of stigma around pain and disability Negative work culture around wellbeing Managers lack understanding about pain and disability Dismissive attitudes among managers Prohibitive management behaviours (preventing or delaying support) Failure to act on OH recommendations Managers disputing OH outcomes

Note: ^+^ SMEs: Small to Medium Enterprises with fewer than 250 employees.

## Data Availability

The data presented in this study are available on request from the corresponding author.

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
