# Peer review of "The Pain at Work Toolkit for Employees with Chronic or Persistent Pain: A Collaborative-Participatory Study"

_healthcare, 2021, doi:10.3390/healthcare10010056_

Round 1

Reviewer 1 Report

This interesting manuscript describes the steps taken to develop the Pain at Work (PAW) toolkit for employees with chronic pain. This is a very important area of work, as there is very much a lack of interventions designed to assist with work-related chronic pain issues. The authors describe four steps, including stakeholder consultation, conducting online surveys with employees and employers, and developing the toolkit.

I commend the authors for including multiple stakeholders, particularly human resource management and those with lived experience of chronic pain. I also appreciate the thorough, yet concise explanation of each step and the corresponding results. Finally, I very much appreciate that the PAW toolkit is publicly accessible and may be widely accessed.

I have a few questions and suggestions for the authors:

  1. Do you have data on race/ethnicity you could add to the demographic presentation in Table 1? Currently only gender, age, and work status are reported. This information is also missing from Table 3.
  2. It is interesting that the 35-54 age group was the most represented for people living with chronic pain. Who do you think this is?
  3. 6, line 239 – missing “to” (regards to)
  4. 9 line 319-320 “recommendation” should be “recommended”
  5. As this study was conducted in the UK, I wonder what cultural differences you think may exist in challenges and/or solutions – particularly for countries with less of a social safety net. I’d like to see you comment on this in the Discussion.
  6. Some of the issues that were brought up by stakeholders could not be addressed in this intervention by nature, as it is meant for employees with chronic pain (and not managers or organizational leaders). Can you add a bit to the Discussion about what types of interventions may be used to complement this one to address some of the more managerial and organizational issues (e.g., decreasing stigma, increasing psychological safety)? It would be helpful to reference studies of such organizational interventions. To this end, I appreciate your paragraph on p. 17 about training – but are there any models or interventions in the literature you can draw from and cite here for people who may be interested in learning more about these?

Thanks for the opportunity to read and comment on your manuscript.

Author Response

Thank you for taking the time to review this manuscript, we appreciate this. Please find our responses uploaded - we believe these have improved the quality of the manuscript. We look forward to hearing from you.

Kind regards

The authors.

Reviewer 2 Report

Line 2: Abbreviations should not appear in the title. In the case that it would be necessary, the explanation of the abbreviation should appear first and then the abbreviation between brackets. When defined for the first time, the acronym/abbreviation/initialism should be added in parentheses after the written-out form.

Line 3: Information about the type of study is missing.

Line 5: The PubMed/MEDLINE standard format is used for affiliations: complete address information including city, zip code, state/province, and country.

Line 9: The abstract should be a total of about 200 words maximum.

Line 11: Again, the explanation of the abbreviation should appear first and then the abbreviation between brackets.

Line 27: Keywords such as toolkit or survey are missing.

Line 35: an explanation of why COVID-19 exacerbates chronic pain is missing.

Line 36: It has been previously described that United Kingdom is UK, therefore is not necessary to repeat that information.

Lines 56 and 58: Why are the references between brackets next to e.g.?

Introduction: Why is necessary this research? A deeper explanation of why an online toolkit is necessary for patients with chronic pain is required. With the present explanation of the manuscript is not justified enough.

Line 79: After reading the manuscript title, I expected that the 3rd objective was going to be the 1st one.

Methods:

Where did the study carry out?

Did an expert in qualitative research carry out the interviews? Which technique was used to analyse the transcriptions?

Further information about the interviews analysis is required.

Line 99: The explanation of the abbreviation should appear first and then the abbreviation between brackets!

Line 119: What was the profession of the included participants?

Results:

Line 194: Characteristics about the type of pain are required (chronic diseases?). I consider that this information could help with a better understanding of the study.

From line 199: Each script should clarify between brackets characteristics of the participant who claimed that, i.e. giving details about gender and age range.

Table 3: I considered that the relevant percentages are the ones described in the total data. The rest are confusing, especially in the "current provision information".

From line 382: Each script should clarify between brackets characteristics of the participant who claimed that, i.e. giving details about size.

Line 573: Do you mean references [4,5,86]?

Line 598: ethical consideration should be available in the methods section.

Limitations of the study?

Author Response

(The authors gave the same response as above.)

Round 2

Reviewer 2 Report

Thank you for your response, that was very helpful to have a better understanding of the research.

Kind regards.